# Development of Mi^a^ Phenotyping Using Paper-Based Device

**DOI:** 10.3390/diagnostics12123104

**Published:** 2022-12-09

**Authors:** Sirinart Chomean, Jirapat Attapong, Sumittra Jitsuvantaya, Komin Poomsaard, Chadchadaporn Dongwilai, Pished Bunnun, Chollanot Kaset

**Affiliations:** 1Department of Medical Technology, Faculty of Allied Health Sciences, Thammasat University, Pathumthani 12120, Thailand; 2Thammasat University Research Unit in Medical Technology and Precision Medicine Innovation, Pathumthani 12120, Thailand; 3Industrial IoT and Automation Research Group (IIARG), National Electronics and Computer Technology Center (NECTEC), 112 Phaholyothin Road, Khlong Luang District, Pathumthani 12120, Thailand

**Keywords:** Miltenberger, paper-based device, HTR, *GYP(B-A-B)* hybrids

## Abstract

The MNS7 (Mi^a^) blood group antigen is found at a different prevalence among different ethnic groups. Anti-Mi^a^ can cause hemolytic disease of the fetus and newborn (HDFN) and both acute- and delayed-type hemolytic transfusion reactions (HTR). Mi^a^ typing should be performed in donors to prevent life-threatening hemolytic transfusion reactions. The gel card and standard tube methods still need specialized equipment, centrifugation, and expertise for result interpretation. We used a novel paper-based analytical device (PAD) pre-coated with monoclonal IgM anti-Mi^a^ for Mi^a^ phenotyping. We measured grey pixel intensity in blood typing results for interpretation processing using OpenCV at the sample (SP) and elution parts (EP); furthermore, we used the SP: EP ratio and F-score as analysis criteria. We typed 214 blood EDTA samples with PAD–Mi^a^ and then compared with gel card results for setting an analysis criterion. We observed 100% sensitivity, specificity, and accuracy when we applied the SP: EP ratio and F-score with the optimal criterion (1.07 and 0.17 for SP: EP ratio and F-score, respectively). The validation of PAD–Mi^a^ typing for blood donor samples (*n* = 150) via F-score gave 100% sensitivity and specificity when compared with the gel card method; therefore, we argue that PAD–Mi^a^ typing can be used for Mi^a^ phenotyping without sero-centrifugation. Moreover, to study the correlation between genotype and phenotype, PCR-SSP was performed to identify *GYP(B-A-B)* hybrids. The results revealed that all Mi^a^+ blood samples gave a positive with GP. Hut, GP. HF, GP. Mur, GP. Hop, and GP. Bun. Results of the gel card method and PCR-SSP were concordant. Hence, using PAD–Mi^a^ typing in blood donors would be helpful for creating a phenotype database of blood donors for reducing alloimmunization risks.

## 1. Introduction

The Mi^a^ (MNS7) blood-group antigen [1] belongs to the low-frequency MNS blood group antigens. Recently, eight hybrid glycophorins express Mi^a^, namely: GP.Vw; GP. Hut; GP. Mur; GP. Hop; GP. Bun; GP.HF; GP. Kip; and GP.MOT [2]. Gene conversion and unequal crossing-over are genetic mechanisms associated with the MNS blood group system. GP. Mur gene conversion enhanced Mi^a^, Mur, MUT, Hut, and MINY expression. Conversely, the GP. Mur homozygote disturbed normal GPB expression. Mi^a^ hybrid glycophorin showed different frequencies among different races, and they are clinically significant because many cases revealed that antibodies against those antigens can cause hemolytic disease of the fetus and newborn (HDFN) and both acute- and delayed-type hemolytic transfusion reactions (HTR) [3,4,5]. The distribution of these glycophorins varies between ethnic populations. The Mi^a^ antigen is quite different when comparing Thai populations—central Thais (9.7%) [6], southern Thais (4.7%) [7], and northeastern Thais (17.9%) [8].

Avoiding alloimmunization for other Rh blood group antigens (D, C, c, E, or e) is routinely performed in European systems for women and patients receiving a chronic transfusion, patients with thalassemia, and sickle cell anemia [9]. In Thailand, the Clinical Practice Guidelines for the management of thalassemia syndromes define the minimal requirement for red blood cell (RBC) antigen matching, including Rh and MNS blood group systems for transfusion to prevent alloimmunization [8].

The tube test and column agglutination test (CAT) are the most common methods for phenotyping blood groups, and they require centrifugation, which is only available under laboratory conditions [10]. Moreover, the process requires trained personnel and is expensive and time-consuming.

The advantages of PADs for blood group typing are that they are cheap, fast, and simple to use [11]. Basically, a blood sample is dropped onto paper treated with antibodies, after which the RBCs react with the corresponding antibodies and then aggregate in the paper, indicating a positive result. Conversely, non-agglutinated blood can flow along the paper after applying a washing buffer, which indicates a negative result [10,12,13,14,15,16]. Of the 11 clinically secondary blood groups that were typed using a PAD (Kleenex paper towel), the results obtained by paper-based devices showed a 100% concordance with the gel card method [10,13]. Most blood groups with antibodies available as IgM, such as C, c, E, e, K and k, Jk^a^ and Jk^b^, and P1, were successfully typed using a PAD, whereas IgG antibodies were unable to cause direct agglutination on a PAD [10].

For several years, it has been recognized that anti-CD38 monoclonal antibodies interfere with pre-transfusion testing by binding to CD38 on red blood cells and causing pan-agglutination on the indirect antiglobulin test (IAT) [17]. This can lead to unwanted testing and significant delays in patient care. Extended phenotyping and genotyping of patient RBCs are helpful for safely providing compatible blood. However, phenotyping must be performed prior to initiation of anti-CD38 antibody therapy and in the absence of both RBC transfusion in the prior 3 months and a positive direct agglutination test (DAT). Blood group genotyping is one approach for overcoming anti-CD-38 interference. Genotyping for RBC antigens can be performed at any time during therapy and can provide more comprehensive detail than phenotyping, particularly regarding minor antigens [17]. In this study, PCR-SSP of *GYP(B-A-B)* hybrids was performed to analyze the correlation of genotype and phenotype outcomes. Thus, the use of PCR-SSP to predict the Mi^a^ antigen in this situation would be helpful to reduce alloimmunization risks.

Our study is the first to use the PAD method for Mi^a^ blood-group typing with four analysis criteria. Thus, paper-based methods for rapid and low-cost Mi^a^ blood group typing are necessary to provide antigen-negative compatible blood and to reduce alloimmunization risks.

## 2. Materials and Methods

### 2.1. Study Design

We designed our study to develop a paper-based Mi^a^ blood grouping. We designed and pre-coated the fabricated paper and cassette with monoclonal IgM anti-Mi^a^. We measured 214 captured pictures (EDTA blood volunteer samples) for grey pixel intensity in our interpretation processing using OpenCV at the sample (SP) and elution parts (EP). In addition, we used the pixel intensity values of SP, EP, SP: EP ratio, and F-score for our analysis criteria. Then, we used 150 EDTA blood donor samples for validation of the analysis criteria. To study phenotype-genotype correlation, PCR-SPP was performed and used to compare with the Mi^a^ phenotype result.

### 2.2. Chemical and Material Used

We purchased a monoclonal IgM anti-Mi^a^ (titer with Mi^a^ cell = 2048) from the National Blood Centre Thai Red Cross Society, Thailand. We used panel cells (known RBC antigen) before their expiration date.

We obtained the ID-diluent II from Bio-Rad Medical Diagnostics GmbH (Dreieich Germany). We used trehalose and sucrose (Sigma-Aldrich, Inc, St. Louis, MO, USA) to stabilize the antibodies on glass fiber. We purchased glass fiber and backing cards from Millipore (Boston, MA, USA). We purchased Pierce Western blotting filter from Thermo Fisher Scientific (Waltham, MA, USA). We obtained CF4 and CF7 papers from GE Healthcare (Little Chalfont, PA, USA).

Our study was approved by the Human Research Ethics Committee of Thammasat University (Science) (HREC-TUSc) (COA No. 083/2562) and performed in accordance with the Declaration of Helsinki and the International Ethical Guidelines for Biomedical Research Involving Human Subjects. We obtained EDTA blood volunteers and the remaining EDTA blood donors after 7 days from the blood bank unit at the Thammasat Hospital University (THU) according to the laboratory protocol. We typed donor blood groups using Ortho Vision Max (Ortho Clinical Diagnostics, Pencoed, UK), which we used as comparative data in our study.

### 2.3. Fabrication of Paper-Based and 3D-Printed Cassette

We designed the cassette to cover the PAD using SOLIDWORKS (DS SolidWorks Corp., Waltham, MA, USA) (Figure 1A). It was rectangular and 325 × 80 × 5 mm in size (Figure 1A). The cassette was composed of a top and a bottom side. The top side provided 2 square holes in the sample section, 2 circular holes 13.0 mm in diameter in the buffer section, and an elution section. We designed the top lid and circular holes at the buffer part to apply the elution buffer to wash non-agglutinated RBCs from the SP where the reaction occurred. We used the square holes in the sample to obtain blood samples. We used the letters “Mi^a^” to demonstrate the lanes for Mi^a^ blood-group antigen detection; furthermore, we used “Ctl” as a systemic control that should not indicate RBC agglutination. We used two channels in the bottom part for the placement of the test and control PADs.

The Mi^a^ blood group PAD consisted of one coated antibody and one control PAD (Figure 1B). We used CF4 paper, which we cut into pieces (13.0 mm in length and 5.0 mm in width), to receive the buffer solution (buffer part). Meanwhile, we designed SP to introduce EDTA blood samples and coated antibodies using glass fibers, which were 12.0 by 5.0 mm in size. For our test PAD, we coated SP with 10 μL of anti-Mi^a^ and 10 μL of preservative solution (5% sucrose and 5% trehalose). For our control system, we only coated the control PAD with a preservative solution on SP. Hence, RBC agglutination would not be seen except for instances of strong direct agglutination (DAT positive). Then, we dried the SP at 37 °C for 2 h. We cut the Pierce Western blotting paper into 34.0 by 5.0 mm pieces, which we used as a flow part after agglutinated or non-agglutinated reactions. We cut the final part of the PAD (CF7 paper) into 9.0 by 5.0 mm pieces, which we designed for EP. We fabricated all parts of the PAD to the plastic backing card (Figure 1C).

All components of the Mi^a^ blood group-typing device are shown in Figure 1D. We printed the designed cassette using a Flashforge Model Creator (Flashforge USA, City of Industry, CA, USA).

### 2.4. Analysis Criteria

We collected 214 EDTA blood volunteer samples to identify the Mi^a^ blood groups. We analyzed all samples in the blood banking laboratory at the THU using the gel card method. We applied an optimized EDTA blood volume (5 μL) on the SP (Appendix A), which we pre-coated with monoclonal IgM anti-Mi^a^. We applied 7 drops of buffer (modified ID-diluent II containing 0.1%Tween 20) to the buffer part after 5 min of reaction time (Appendix A) to elute a non-agglutinated cell flow at the end of the PADs (EP). We captured blood group typing results using Oppo A5 2020 smartphone (Android version 10, Qualcomm^®^ Snapdragon™665 and RAM 3.0GB) to analyze the result. Then, we measured the grey pixel intensity of the agglutinated (at SP) and non-agglutinated cells (EP) using OpenCV.

Blood group typing should be clearly differentiated between positive and negative results to prevent false-negative and false-positive results. Due to immunogenicity and the type of antibody we used, our analysis of pixel intensity at the SP and EP was insufficient to interpret minor blood-group typing. We developed analysis criteria in our study and combined them with a specific formula, including the SP: EP ratio and F-score. We computed the grey pixel intensity values to create the F-score using:F = (SPt/EPt) − (SPc/EPc)

### 2.5. Validation of Mi^a^ Typing Using Real Samples

The 150 EDTA blood donor samples were used to validate the Mi^a^ PAD and image processing. Blood samples (5 μL) were applied to SP. After 5 min, 7 drops of modified ID-diluent II were applied on the buffer part, and the results were captured and imported into OpenCV using an optimal threshold.

### 2.6. PCR-SSP for Mi^a^ Genotyping

Genomic DNA was extracted from 150 EDTA donor blood samples (from Section 2.5) using the Gentra Puregene Kits according to the manufacturer’s instructions (QIAGEN GmbH, Valencia, CA, USA), then stored at −20 °C until used. Quantity of genomic DNA was performed using a NanoDrop™ One/OneC Microvolume UV-Vis Spectrophotometer (Thermo Fisher Scientific, USA). A PCR-SSP technique was performed to detect two MNS hybrid GPs according to a previous study [18]: set I—GP. Hut, GP. HF, GP. Mur, GP. Hop, and GP. Bun (F2 primer (5′-CCCTTTCTCAACTTCTCTTATATGCAGATAA-3′) and Rccgg primer (5′-GAGCAACTATTTAAAACTAAGAACA TACCGG-3′); set II—GP.Vw (5′-CAG-CATTTCTCTAAAGGCTAAATAAGAAGATG-TA-3′) and RIN primer (5′-CATATGTGTCCCGTTTGTGCA-3′). In the total 12.5 μL reaction, 2 μL of genomic DNA (50 ng/μL) was amplified using 1 μL of 10 μM F2 primer and 1 μL of 10 μM Rccgg primer for set I. For the second set, 1 μL of 10 μM F1 primer and 1 μL of 10 μM RIN primer (5′-CATATGTGTCCCGTTTGTGCA-3′) was used. In addition, 1μL of 5 μM HGH-434-F primer (5′-TGCCTTCCCAACCATTCCCTT A-3′) and 1 μL, 5μM of HGH-434-R (5′-CCACTCACGGATTTCTGTTGTGTTTC-3′) primer were included in the PCR reaction as an internal control gene. A PCR-SSP was performed in a T100 thermal cycler (Bio-Rad Laboratories, Inc., Hercules, CA, USA). The cycling parameters for the PCR program consisted of 1 cycle at 95 °C for 15 min, followed by 34 cycles at 95 °C for 20 S, 62 °C for 20 S, and 72 °C for 20 S, with a final extension at 72 °C for 10 min. PCR amplicon was separated in 1.5% agarose gel with 80V 400 mA for 60 min which was stained with 10,000X SYBR (Invitrogen, Waltham, MA, USA). The result was visualized and captured using a Blue light LED transilluminator.

### 2.7. Statistical Analysis

We analyzed the statistical significance using Statistical Package for the Social Sciences (SPSS) for Windows, version 18.0, and MedCalc Statistical Software, version 20.111 (MedCalc Software bv, Ostend, Belgium). We conducted Mann–Whitney tests for non-parametric data and independent sample comparisons between positive and negative results. We conducted ROC curve analysis to determine the threshold values for separating the positive and negative results. We used AUC analysis to evaluate the applied performance. Finally, we analyzed the performance of the Mi^a^ PAD system using the % specificity, % sensitivity, % accuracy, % Positive Predictive Value (PPV), and % Negative Predictive Value (NPV). We considered a statistical *p*-value < 0.05 as statistically significant.

## 3. Results

### 3.1. Analysis Criteria

Based on the agglutination reaction, a specific antibody reacts with the corresponding antigen on the RBC in the test tube, which can be observed after centrifugation and interpreted by the naked eye. Unlike ABO blood group typing, minor blood group typing requires only type antigens on RBC, and several factors affect the interactions between antigens and antibodies. For this reason, the interpretation of EP and SP may not be reliable for minor blood group typing. In our study, the Mi^a^ blood group phenotyping (Figure 2A,B) PADs and cassette designs are shown in Figure 1A–C. We processed the images to grey scale and analyzed the color intensities in the SP and EP areas responsible for SP_T_, EP_T_, SP_C_, and EP_C_ using OpenCV (Figure 1D). The pixel intensity of the EP and SP was transformed into a fraction (F-score) as (SP_T_ ÷ EP_T_) sample–control (SP_C_ ÷ EP_C_). We calculated the cut-off for each algorithm using MedCalc Statistical Software. A higher value (>ut-off, 0.17) was positive, whereas a lower value (≤cut-off, 0.17) indicated a negative value. (Appendix A).

Overall, the pixel intensities of SP, EP, SP: EP ratio, and F-score were significantly different between positive and negative results (*p* < 0.0001) (Figure 3A, Figure 3B, Figure 3C and Figure 3D, respectively). However, a positive/negative overlap was observed for the pixel intensities of SP, EP, and SP: EP ratio (Figure 3A, Figure 3B, and Figure 3C, respectively).

We performed ROC curve analysis using MedCalc to set the cut-off. The SP pixel intensity revealed AUC = 0.980 (*p* < 0.0001: 95%CI = 0.951–0.994) (Figure 4A), and the optimal criterion was more than 160.17, which gave 62.96% sensitivity, 100% specificity, and 96.41% accuracy (Table 1). Likewise, the pixel intensity from EP showed AUC = 0.988 (*p* < 0.0001: 95%CI = 0.962–0.998) (Figure 4B), and the optimal criterion was less than or equal to 126.73, which gave 96.30% sensitivity and 100% specificity (Table 1). We observed 100% sensitivity, specificity, and accuracy (Table 1) when we applied the SP: EP ratio and F-score (*p* < 0.0001: 95%CI = 0.983–1.000) (Figure 4C and Figure 4D, respectively) with the optimal criterion (1.07 and 0.17 for SP: EP ratio and F-score, respectively).

### 3.2. Validation of Mi^a^ Typing Using Real Samples

An optimal cut-off (F-score = 0.17) was used as a criterion for Mi^a^ typing interpretation. The 150 EDTA blood donor samples were subjected to blood typing using PAD. Blood typing results were captured and imported into OpenCV. A higher F-score (>0.17) indicated positive, while a lower F-score (≤0.17) indicated negative. The Mi^a^ typing with F-score were given 100% sensitivity, specificity, accuracy, PPV, and NPV. We observed the Mi^a^ antigen in 12% (18/150) of our study population.

### 3.3. Phenotype-Genotype Correlation Using PCR-SSP

The PCR-SSP results of MNS hybrid GP detections are shown in Figure 5. The *GYP(B-A-B)* hybrids of GP. Hut, GP. HF, GP. Mur, GP. Hop, and GP. Bun were amplified with amplicon size of 148 or 151 bp and 434 bp for internal control gene (HGH).

Of the 150 EDTA blood donors, 12% (18/150) were positive with the set of primers specific for GP. Hut, GP. HF, GP. Mur, GP. Hop, and GP. Bun. No GP.Vw was detected by the PCR- SSP technique (data not shown).

## 4. Discussion

This study aims to develop the PAD method for Mi^a^ blood group typing with four analysis criteria based on image-based analysis, including: SP; EP; SP: EP ratio; and F-score. The Mi^a^ phenotyping with F-score were given 100% sensitivity, specificity, accuracy, PPV, and NPV. Moreover, PCR-SSP of *GYP(B-A-B)* hybrids was performed to observe the correlation of genotype and phenotype. Among the 150 EDTA blood donor samples, only GP. Hut, GP. HF, GP. Mur, GP. Hop, and GP. Bun were detected.

Pre-transfusion tests consisted of ABO and RhD typing a patient’s red blood cells and an unexpected antibody screen with the patient’s plasma. However, antibodies to non-ABO antigens (alloantibodies) often develop in multi-transfused patients exposed to RBC antigens different from their own. Phenotyping Rh (C, c, E, and e) and Mi^a^ is required for matching donors and recipients to reduce post-transfusion reactions [8,19,20,21]. Technically, serological blood typing still requires a longer turnaround time and is more expensive (column agglutination method). Our study aims to design and develop a PAD- and image-based analysis criteria method, which could be used as an alternative method for Mi^a^ phenotyping. Although PAD technologies and microfluidic platforms can produce a direct readout of clinical blood samples without sero-centrifugation, well-trained technicians are still needed when interpreting blood group results.

Unlike the ABO and RhD, we noticed variations of agglutination grading when minor blood group antigens reacted with specific antibodies. The level of agglutination can be affected by the immunogenicity of blood group antigens, antibody titers, and reaction times [10,12]. Wei et.al. stated that a short reaction time (10–30 sec) is not sufficient for some minor blood group antigens because they are less antigenic, for example Rh (D > c > E > C > e) [10,12]. As a result, the time required for minor blood group agglutinations interpretation ranges from 30 s to 3min, depending on the RBC antigen–antibody. Our study results show that a reaction time at 5 min is appropriate for Mi^a^ blood typing interpretation (Appendix A). However, the interpretation of EP and SP may not be reliable for minor blood group typing [10,21]. In our study, we transformed the pixel intensities of EP and SP into a fraction (SP: EP ratio and F-score). Figure 4 shows that F-score could be used as a candidate algorithm because of this formula, which was calculated by subtracting the control baseline. Consistent with previous studies, W. L. Then et al. argued that the intensity of a blood spot (BS) alone was insufficient to indicate a reaction between an RBC antigen and antibody [10]. Therefore, we measured an area of the EP above the BS in addition to the BS intensity. We transformed this difference into a fraction, and we then used the optical density ratio (ODR) as analysis criteria for minor blood group interpretation [10]. Overall, we observed the Mi^a^ antigen in 12% (18/150) of our study population and revealed a significant difference in the northern Thai (17.97%) and southern Thai (4.7%) populations (*p* < 0.001) [2,7].

Compared with the tube method, the gel card and molecular methods are more expensive, take longer, and are not suitable for every blood bank unit [11,12,15]. However, our developed PAD can be used as an alternative method for Mi^a^ grouping, even in routine or emergency situations. Nevertheless, we observed several limitations in our proposed method. First, anticoagulated samples from patients (not transfused within 3 months prior to study) and newborns (cord blood) should be evaluated to ensure the effectiveness of our proposed method. Second, the number of complex samples should be included: reactive DAT-positive samples collected from patients and ‘mixed-field’ samples (two erythrocyte populations) obtained from patients who had been transfused within a few days to 3 months prior to the study require further observation [22,23].

However, some monoclonal antibody therapy can cause interference on pre-transfusion testing. Anti-CD38 (daratumumab) interferes in the indirect antiglobulin testing, whereas anti-CD47 interferes with all phases of pre-transfusion testing [9,17]. The genotyping could be performed in this situation [17]. In this study, all Mi^a^ positive samples (18/150) were positive with primer F2, Rccgg, which detect GP. Hut, GP. HF, GP. Mur, GP. Hop, and GP. Bun (Figure 5). Results of genotyping 18 Mi^a^ + and 132 Mi^a^ − by gel card method and PCR-SSP were concordant. Similarly, Palacajornsuk and colleagues reported that 29 Mi^a^ + (29/59) DNA samples were positive with GP. Hut, GP. HF, GP. Mur, GP. Hop, and GP. Bun [18]. Nathalang et al. also found that all 94 Mi^a^+ samples were positive for MNS hybrids (83 were GP. Mur and 11 were GP. Bun, confirmed by DNA sequencing) and GP.Vw was not found by PCR-SSP [7]. By PCR and sequencing, 25 from 389 DNA blood samples were proved to be GP. Mur genotype [24]. Of the 11 Mi^a^ positive (11/5098) MNS hybrid identified, GP. Mur was the most common type in the Australian blood donor population [1]. It can be concluded that GP. Mur was the most common phenotype in Asian and Australian populations. The Mi^a^ + genotype among Thai blood donors (12%) revealed that Mi^a^ + frequency is quite varied even among the Thai population, with it being found in 22.3% in the northern Thai population [8] and only 4.7% in the southern Thai population [7]. Based on the two primer sets used [18], PCR-SSP with the first set of primer was performed to detect hybrid glycophorins GP. Hut, GP. Mur, GP. Bun, GP. Hop, and GP. HF, but it does not specify the type of hybrid glycophorins. GP. Mur is commonly found in Asian and Australian populations [1,18,25,26,27]. GP. Mur homozygous individuals do not express normal GPB (JENU) which can produce antibodies to normal GPB. Thus, DNA sequencing is needed to define the specific hybrid glycophorin including GP.Kipp and GP.MOT [28,29].

The developed PAD can be used as an alternative method for Mi^a^ grouping. With specific analysis criteria, the proposed method may be developed into an automated portable platform. Moreover, 100% concordance was observed between the phenotype and genotype results. Therefore, PCR-SSP can predict for Mi^a^ antigens in cases where the phenotyping of blood groups could not be tested.

## 5. Conclusions

The PAD we developed can be used as an alternative method for grouping Mi^a^ or other minor blood-group antigens. Using specific analysis criteria, our proposed method may be developed into an automated portable platform, which would be useful for time- and resource-limited environments.

## Figures and Tables

**Figure 1 diagnostics-12-03104-f001:**
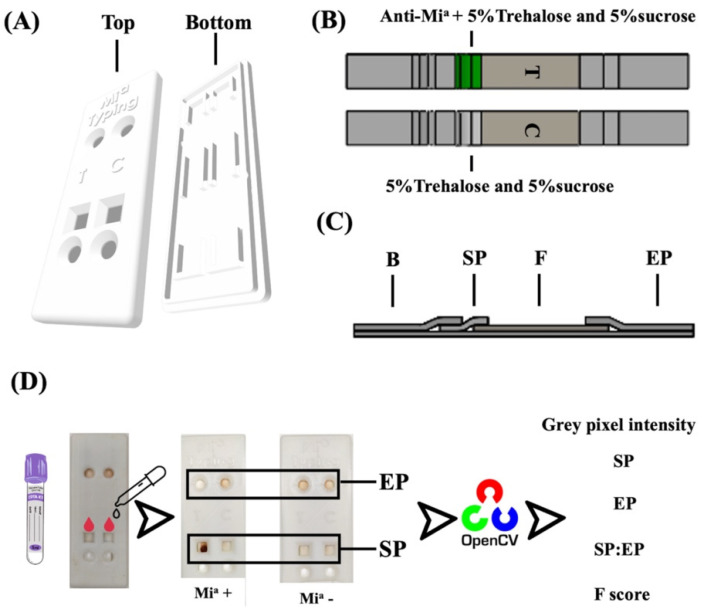
Illustration of minor blood group phenotyping concept: (**A**) top and bottom 3D cassette and PAD; (**B**) PAD–Mi^a^ phenotyping for test and control line; (**C**) composition of PAD–Mi^a^ phenotyping: B: buffer part; SP: sample part; F: flow part; and EP: elution part; (**D**) image processing for creating analysis criteria.

**Figure 2 diagnostics-12-03104-f002:**
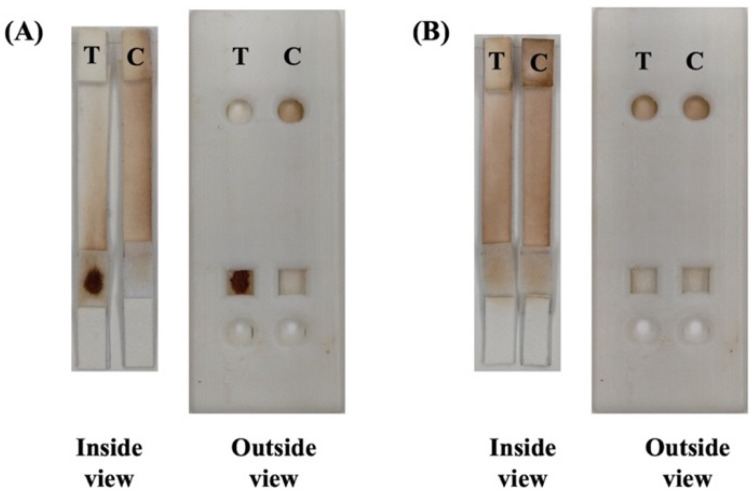
Inside and outside views of PAD for Mi^a^ blood group typing (**A**) Mi^a^ positive; (**B**) Mi^a^ negative.

**Figure 3 diagnostics-12-03104-f003:**
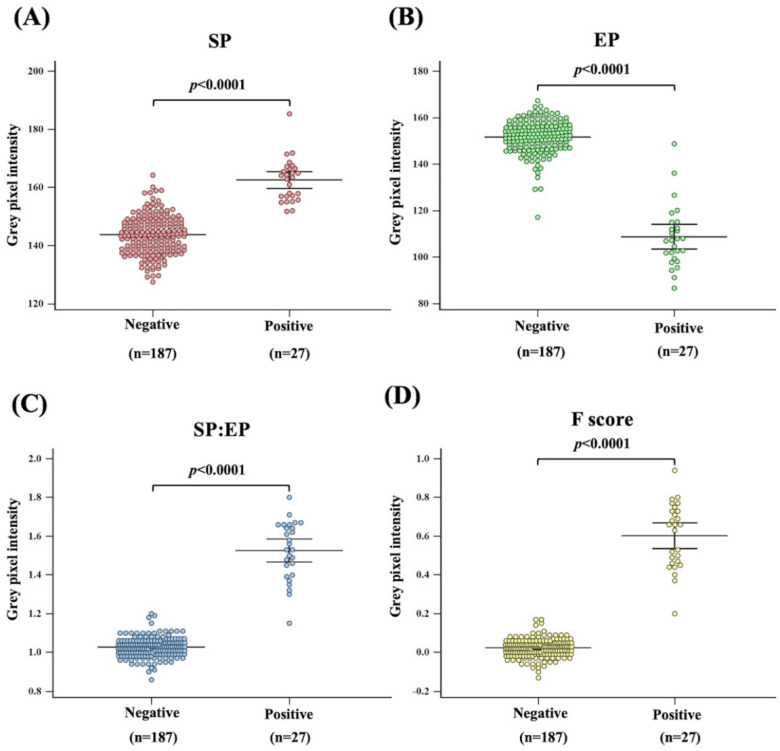
Comparison of grey pixel intensity between Mi^a^ negative and Mi^a^ positive using Mann–Whitney tests. (*p*-value < 0.05 as statistically significant.). Grey pixel intensity in different analysis criteria observed by (**A**) SP; (**B**) EP; (**C**) SP:EP ratio and (**D**) F score.

**Figure 4 diagnostics-12-03104-f004:**
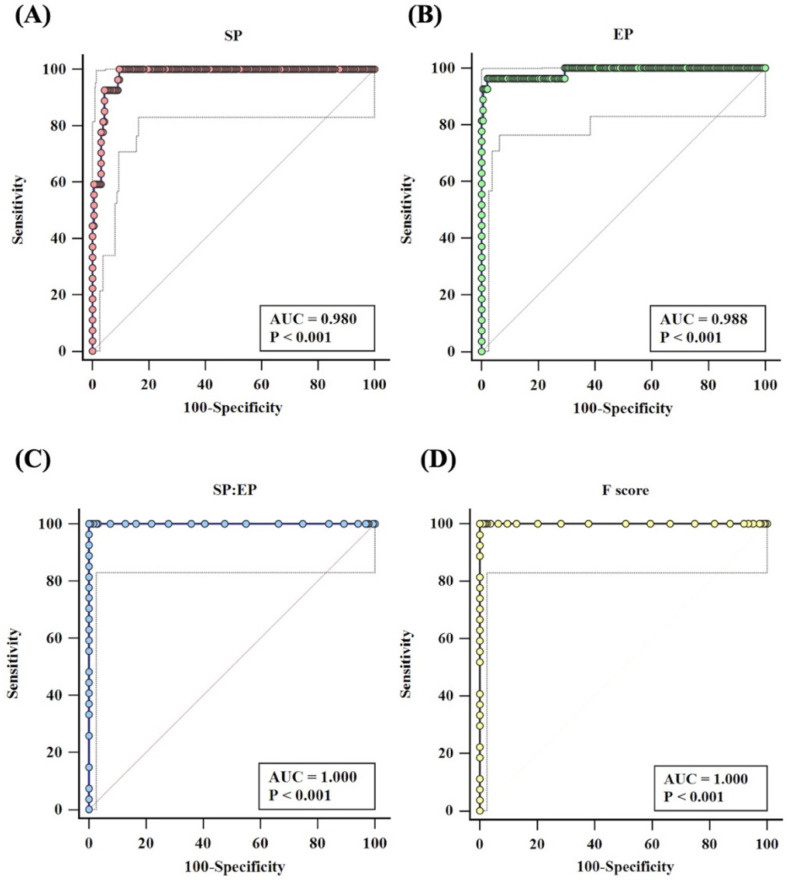
ROC curve analysis of grey pixel intensity from (**A**) SP, (**B**) EP, (**C**) SP: EP ratio, and (**D**) F-score.

**Figure 5 diagnostics-12-03104-f005:**
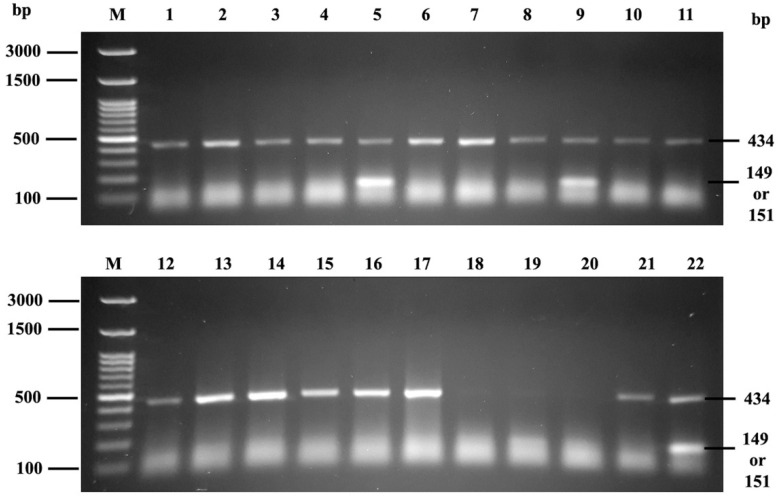
Mi^a^ genotyping by PCR-SSP. Lane M: 100 ladder (Biotech rabbit, Germany), Lane 1–4, 6–8, 10–17: Mi^a^ negative. Lane 18–19: Poor DNA template. Lane 5, 9: Mi^a^ Positive. Lane 20: Non template DNA (NTC). Lane 21: Negative control (NC) and Lane 22: Positive control (PC).

**Table 1 diagnostics-12-03104-t001:** Performance of PAD–Mi^a^ using four different analysis criteria (SP, EP, SP: EP ratio, and F-score).

STATISTICS	SP		EP		SP: EP Ratio		F-SCORE	
	Value	95%CI	Value	95%CI	Value	95%CI	Value	95%CI
Sensitivity	62.96%	42.37% to 80.60%	96.30%	81.03% to 99.91%	100.00%	87.23% to 100.00%	100.00%	87.23% to 100.00%
Specificity	100.00%	98.05% to 100.00%	100.00%	98.06% to 100.00%	100.00%	98.05% to 100.00%	100.00%	98.05% to 100.00%
PPV	100.00%	-	100.00%	-	100.00%	-	100.00%	-
NPV	96.17%	93.89% to 97.62%	99.60%	97.35% to 99.94%	100.00%	-	100.00%	-
Accuracy	96.41%	92.96% to 98.47%	99.64%	97.62% to 100.00%	100.00%	98.29% to 100.00%	100.00%	98.29% to 100.00%

## Data Availability

Not applicable.

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
