# Peer review of "Development of Mia Phenotyping Using Paper-Based Device"

_diagnostics, 2022, doi:10.3390/diagnostics12123104_

Round 1
Reviewer 1 Report
Comments to the Authors
The manuscript (Manuscript ID: diagnostics-2000692) by Chomean S. et al. described a blood group phenotyping method for Mi(a) (MNS7) using a paper-based device (PAD) pre-coated with an anti-Mi(a) IgM monoclonal antibody.
The optimisation assay was performed by comparing 214 captured pictures (from 214 EDTA-blood samples) with phenotyping results from gel cards. Pixel intensity values were obtained from PAD pictures. SP:EP ratio and F-score values were used to assess presence (positive) or absence (negative) of Mi(a) on blood samples. For the validation assay, PAD phenotyping results from 150 EDTA blood donor samples were compared with PCR-SSP genotyping results. PAD Mi(a) phenotyping showed 100% specificity and specificity. Manuscript was well-written.
Comments
1. Line 27: The use of the “Miltenberger subsystem” terminology is obsolete. They can be referred as MNS hybrid glycophorins or MNS hybrid glycophorin variants. Only six Mi(a)-positive hybrid glycophorins were mentioned in this manuscript. Currently, there are eight including GP.Kip and GP.MOT. Please give a more up-to-date Introduction.
2. Line 35: In European countries and in other countries with a significant D-negative population, preventing alloimmunisation to D antigen is also important in addition to C, E, c, and e antigens.
3. Line 56. The authors stated that the PAD method is low-cost and fast. Do you have information as to how much cheaper and quicker PAD Mi(a) phenotyping is to perform compared to tube method and gel card typing?
4. Line 61-62. Were PADs coated with anti-Mi(a) typing reagent at 128-titre or was the reagent further diluted? If information is available, describe how strong the agglutination reaction of a 128-titre typing reagent with Mi(a)-positive RBCs on gel cards/tube method.
5. Line 70. Describe the name of the anti-Mi(a) IgM clone used in this study? Is this typing reagent a single clone or is this a pool of anti-Mi(a) monoclonal antibodies? Is there an article or reference to show that this anti-Mi(a) IgM clone can detect all or only some of the Mi(a)-positive hybrid glycophorins?
6. Line 115-116. This study used blood donor samples collected in EDTA tubes (assuming 6mL EDTA-tubes). Figure 1D shows a finger with a blood drop. In my opinion, this figure gives the impression that a finger-prick blood sample, with no anti-coagulants, can be used for PAD Mi(a) typing which might be misleading. This reviewer suggests removing the “finger with a blood drop” diagram.
7. Line 130: Can smart phones other than Oppo A5 2020 be used to capture pictures for analysis? Or, is it strictly Oppo A5 2020 only?
8. Line 150-156. The authors described a PCR-SSP assay detecting six MNS hybrid glycophorins but did not explain the association of these hybrid glycophorins with the Mi(a) antigen (MNS7) expression. In the Introduction section, consider including background information on MNS hybrid glycophorins that express Mi(a).
9. Line 150-162: If this PCR-SSP assay is from a published article, please cite reference here.
10. Line 190-192: State the cut-off values for negative and positive results.
11. Line 219-220. What is the interpretation if the F score is equal to 0.17?
12. Line 222: Please check the calculation for 12% Mi(a)-positive samples. It should be 18/150. The fraction 18/132 (X 100%) is 13.63%.
13. Line 225: GYP(B-A-B) is a gene must be written in italics
14. Line 228: There were 150 blood donors tested in this validation test. Please explain why the denominator is 214.
15. Line 228-229: Gel-PCR assay using Set 2 primers (F1 and RIN) was performed but result for this investigation was not described.
16. Line 239-241 and 264-265. The authors compared the cost between tube method phenotyping vs gel card vs molecular typing but did not mention the cost for PAD Mi(a) phenotyping. Could the authors give an indication how much PAD Mi(a) phenotyping costs?
17. Line 239-241. The manuscript stated that serological typing is not established in every blood bank unit. Did you mean in Thailand only or is this observed worldwide?
18. Line 266, Line 288, Line 295. The manuscript described that PAD Mi(a) phenotyping can be used in emergency situations. I am assuming that blood bags issued by blood banks would have been phenotyped generally using the tube method or gel cards. Therefore, are the authors suggesting to use PAD Mi(a) phenotyping on patients?
This manuscript describes PAD Mi(a) phenotyping on healthy blood donors only and not on patients. In what emergency situations would Mi(a) phenotyping on patients be warranted? What is the clinical significance? This is may be true if the patient is GP.Mur/Mur homozygote, JENU-negative and has developed anti-JENU antibody. Therefore, blood for transfusion must be matched for ABO and Rh, and must also be GP.Mur/Mur homozygote JENU-negative.
19. Line 267-270. Consider these limitations:
o A) the PCR-SSP assay (F2 and Rccgg) was designed to detect or confirm sequences in hybrid glycophorins that encode Mi(a) but it does not specify the type of hybrid glycophorins whether it is GP.Hut, GP.Mur, GP.Bun, GP.Hop, or GP.HF.
o B) that the PAD Mi(a) phenotyping was not performed on patients (generally unwell) and who may have a range of diseases that may affect PAD Mi(a) typing result. For example, would Mi(a)-positive blood from a patient who has severe anaemia give a false-negative PAD result? Also, newborns and people with polycythaemia generally have higher haemoglobin and RBC count than adults. Therefore, would blood samples from newborns and individuals with polycythaemia give a false-positive PAD result?
20. Line 294-295: Is PAD blood typing technology currently used in the diagnostic setting in Thailand?
Author Response
Thank you for giving me the opportunity to submit a revised draft of my manuscript titled Development of Mia phenotyping using paper-based deviceto Diagnostics. We appreciate the time and effort that you and the reviewers have dedicated to providing your valuable feedback on my manuscript. We are grateful to the reviewers for their insightful comments on our paper. We have been able to incorporate changes to reflect most of the suggestions provided by the reviewers. We have highlighted the changes within the manuscript.
"Please see the attachment"
Here is a point-by-point response to the reviewers’ comments and concerns.

Reviewer 2 Report
In this very interessant study, the authors developed a PAD that can be used as an alternative method to grouping Mia or other minor blood group antigens, even in routine or emergency situations.
Minor comments:
The introduction is a little too short and needs to be better developed by further explaining the hypothesis of the study
Why all samples are taken on EDTA?
Author Response
Thank you for giving me the opportunity to submit a revised draft of my manuscript titled Development of Mia phenotyping using paper-based deviceto Diagnostics. We appreciate the time and effort that you and the reviewers have dedicated to providing your valuable feedback on my manuscript. We are grateful to the reviewers for their insightful comments on our paper. We have been able to incorporate changes to reflect most of the suggestions provided by the reviewers. We have highlighted the changes within the manuscript.
"Please see the attachment"

Reviewer 3 Report
Review on manuscript JCLA-22-1826: ‘Development of Mia phenotyping using paper-based device’ by Sirinart Chomean and colleagues.
General comment:
The authors evaluated the performance of a new paper-based device (PAD), pre-coated with monoclonal anti-Mia for Mia phenotyping. Results from PAD were compared with those from gel results and from Mia genotyping. Results from PAD were in concordance with the two other techniques. The frequency of Mia in the 214 donor samples was 12.6%. The authors concluded that PAD can be used as an alternative method for Mia grouping.
This study is of interest, being the first study using the PAD method for Mia antigen determination.
However, there are comments that need to be addressed.
1. Several publications were incorrectly cited (ref 2, 6, 7, 12-15, etc.) or the original publication was not cited. I did not check all references, but the authors should.
2. It is unclear how many samples were actually tested, 214, 150, 132 or 59.
3. Although the test was accurate with normal hematocrit (i.e. tested using blood from blood donors), the study by Noiphung et al. (ref 12) showed that the hematocrit of the sample affects the accuracy of the test. In an anemic patient Hct is (very) low (emergency situations). I suggest that the authors also establish the performance of their new PAD with abnormal Hct.
4. The authors only tested samples with an incubation time of five minutes, while other studies testing PAD with other blood groups reported shorter incubation time for optimal results. Perhaps PAD Mia determination is already possible after a three minutes (or less) incubation.
Specific comments:
Title: No comment
Abstract:
5. Actually PAD is the abbreviation of paper-based analytical device.
6. It should be clear from the abstract that results were also compared with genotyping.
Introduction: No comments
Materials and methods:
7. Page 2, line 71: What panel cells were used and where were these used for?
8. Lines 77 and 86: These two sentences are exactly the same, also for 89 and 90.
9. Page 4, line 126: What is meant by an optimized EDTA blood sample?
10. Line 135: Unclear what the authors mean by this sentence, in addition give a reference for this statement.
11. Line 141: What is meant by ‘The 150 samples..’, a total of 214 samples were tested in the previous version of this manuscript.
12. Line 145: Write out abbreviations when used for the first time (PPV, NPV)
13. Line 147: State that genomic DNA was extracted from blood from the same donors.
14. Line 171: Where was Mann-Whitney U test used?
15. Page 5, Line 175: This sentence is a repetition of page 4, line 145.
Results:
16. Line 197: When the pixel intensities of SP and EP are already significantly different between positive and negative results for every time point, why calculate SP:EP and F-score? The authors probably mean the mean/median pixel intensities. However, when I interpreted figures S1, than Mia+ results are clearly distinguishable from Mia- results. In addition, what is meant by ‘every time point’?
17. Page 8, line 218: Again 150 instead of 214 samples.
18. Line 222: Here it are 132 samples and 18/132=13.6%. In the previous version the frequency was 27 in 214 samples.
19. Line 223: Unclear where this 17.97% comes from.
20. Line 226: Redundant with M&M section.
Discussion:
21. It is common use to start the discussion with a summary of the main findings. 22. Page 9, line 255: This figure (S1) does not contain information on a reaction time of 5 minutes, only different blood volumes.
23. Line 259: G. Garnier not in the reference list.
24. Line 272: Does this also apply for Mia blood group determination, if so, give a reference.
25. Line 277: Here 59 (not 214?) samples were tested and 29 samples were positive. Actually 27 samples were positive (Fig S2-10).
26. Line 286: Provide references for the 22.3% and 4.7%.
27. Line 295: This was not tested and cannot be concluded from this study.
28. Page 10, line 299: The link does not work.
References: No comments
Tables and figures:
29. What is depicted in fig 3. means (sd), median (range IQR) or other and what analysis was used to compare results.
30. Also, not clear what interesting information is provided by this figure. It shows grey pixel intensity (SP and EP) in Mia+ and Mia- samples, but not the difference between individual samples, which is of more interest/importance.
Author Response

(The authors gave the same response as above.)

Round 2
Reviewer 1 Report
Comments to the Authors
General comment
The revised manuscript (Manuscript ID: diagnostics-2000692_R) by Chomean S. et al. has improved. This reviewer is satisfied with the changes made in the revised manuscript in response to comments made on the original manuscript.
GP.Mur/Mur individuals do not express normal GPB and have been reported to produce anti-s, anti-U, anti-JENU, and more recently, anti-sD. The sD antigen of the MNS blood group system was found in 2.2% of Thai blood donors. Mi(a)+ hybrid glycophorin, such as GP.Mur, is more prevalent in the Thai population compared to other population groups. Therefore, it is important that Mi(a+) individuals are accurately identified. Congratulations on your manuscript.
Minor comments:
1. Line 18, 19, 85, 216, 229, 230, 233, in Table 1, 237: For clarity, ensure “ratio” is written after the word “SP:EP” such as “SP:EP ratio”.
2. Line 46: There is no “Mi blood group system”. Change it to “MNS blood group system”.
3. Line 72 and 260: Ensure that GYP(B-A-B) is written in italics and no spaces in between.
4. Line 252-253: In Figure 5, the specific band produced by F2/Rccgg is 148 bp and not 149bp.
5. Line 310, 312, 313, 319, 320: After the word Mi(a), the plus (+) or minus (–) signs should be written in normal text and not in superscript. Please check throughout the manuscript.
6. Line 323: Please check the spelling. It should be “performed” and not “preformed”.
Author Response
Response to Reviewer 1 Comments
Thank you for giving me the opportunity to submit a revised draft (2nd round) of my manuscript titled Development of Mia phenotyping using paper-based device to Diagnostics. We appreciate the time and effort that you and the reviewers have dedicated to providing your valuable feedback on my manuscript. We are grateful to the reviewers for their insightful comments on our paper. We have been able to incorporate changes to reflect most of the suggestions provided by the reviewers. We have highlighted the changes within the manuscript.
Here is a point-by-point response to the reviewers’ comments and concerns.
Point 1. Line 18, 19, 85, 216, 229, 230, 233, in Table 1, 237: For clarity, ensure “ratio” is written after the word “SP:EP” such as “SP:EP ratio”.
Response 1: Thank you for pointing this out. We agree with this comment. Therefore, We have checked and written after the word “SP:EP”. Mention exactly where in the revised manuscript this change can be found – Line 18,21, 22, 90, 161, 223,226, 236, 238, 240, Table 1, 244, 265, and 294.
Point 2. Line 46: There is no “Mi blood group system”. Change it to “MNS blood group system”.
Response 2: Thank you for pointing this out. Mi blood group system has been changed to “MNS blood group system”. Mention exactly where in the revised manuscript this change can be found – page 2 and line 51.
Point 3. Line 72 and 260: Ensure that GYP(B-A-B) is written in italics and no spaces in between.
Response 3: Thank you for pointing this out. We agree with this comment. Therefore, We have checked and corrected it. Mention exactly where in the revised manuscript this change can be found –line 26, 30, 77, 253-254, 267 and 446.
Point 4. Line 252-253: In Figure 5, the specific band produced by F2/Rccgg is 148 bp and not 149bp.
Response 4: Thank you for pointing this out. We have corrected specific band from 149 to 148 bp Mention exactly where in the revised manuscript this change can be found – page 9 and Figure 5.
Point 5. Line 310, 312, 313, 319, 320: After the word Mi(a), the plus (+) or minus (–) signs should be written in normal text and not in superscript. Please check throughout the manuscript.
Response 5: Thank you for pointing this out. We agree with this comment. We have checked and corrected it. Mention exactly where in the revised manuscript this change can be found – Line 26, 320,322, 329, 330.
Point 6. Line 323: Please check the spelling. It should be “performed” and not “preformed”.
Response 6: Thank you for pointing this out. We agree with this comment. We have corrected it. Mention exactly where in the revised manuscript this change can be found – Line 333.